# Measurements of morphodynamics of a sheltered beach along the Dutch Wadden Sea

Marlies A. van der Lugt[1, 2], Jorn W. Bosma[3], Matthieu A. de Schipper[1], Timothy D. Price[3], Marcel C.G. van Maarseveen[3], Pieter van der Gaag[1], Gerben (B.G.) Ruessink[3], Ad J.H.M. Reniers[1], and Stefan G.J. Aarninkhof[1]

[1]Delft University of Technology, Delft, The Netherlands
[2]Deltares, Delft, The Netherlands
[3]Utrecht University, Utrecht, The Netherlands

**Correspondence:** Marlies van der Lugt (m.a.vanderlugt@tudelft.nl)

**Abstract.** A field campaign was carried out at a sheltered sandy beach with the aim of gaining new insights into the driving processes behind sheltered beach morphodynamics. Detailed measurements of the local hydrodynamics, bed level changes and sediment composition were collected at a man-made beach on the leeside of the barrier island Texel, bordering the Marsdiep basin that is part of the Dutch Wadden Sea. The dataset consists of (1) current, wave and turbidity measurements from a
dense cross-shore array and a 3 km alongshore array, (2) sediment composition data from beach surface samples, (3) high-temporal-resolution RTK-GNSS beach profile measurements, (4) a pre-campaign spatially covering bathy-topo map and (5) meteorological data. This manuscript outlines how these measurements were set up and how the data have been processed, stored and can be accessed. The novelty of this data set lies in the detailed approach to resolve forcing conditions on a sheltered beach, where morphological evolution is governed by a subtle interplay between tidal and wind-driven currents, waves and
bed composition, primarily due to the low-energy (near-threshold) forcing. The data is publicly available at 4TU Centre for Research Data at DOI: 10.4121/19c5676c-9cea-49d0-b7a3-7c627e436541 (van der Lugt et al., 2023).

## 1   Introduction

Sheltered coastlines are protected from high wave impact and can be found in estuaries, coastal basins or inland lakes. Sandy sheltered coastlines typically undergo smaller rates of storm-driven erosion than exposed coasts and received less attention
in research. Nevertheless, understanding the physics of these beaches is just as important, as they also protect vital coastal infrastructure and communities that rely on them as a defense against flooding.

The morphodynamic evolution of sandy beaches in sheltered environments is governed by both waves and currents (Freire et al., 2009). The wave forcing is generally characterised as low-energy: fair weather wave heights are small ( $H_{m0} < \mathcal{O}(0.25)$ m), and storms with onshore wind have maximum wave heights constrained by the geometry of the adjacent (semi-) enclosed
basin ($H_{m0} < \mathcal{O}(1.5)$ m). Beaches in estuaries and on the lee-side of islands may be typically sheltered from long period swell, but under the right conditions some refracted ocean swell waves may still reach these shores (Cooper et al., 2007).

Predicting morphodynamics on sheltered beaches has specific challenges. The beach face is generally narrow ($< \mathcal{O}$ (20) m) and morphological features inherited from storm events can persist for long time periods (Jackson et al., 2002; Nordstrom and Jackson, 2012). In addition, the planform setting of sheltered beaches is often less alongshore uniform as beaches are curved,
enclosed between headlands, channels or structures. Consequently, alongshore gradients and large-scale circulation patterns or residual currents are drivers of typical morphological features of sheltered areas, such as shoals and spits (Hopkins et al., 2017; Van Kouwen et al., 2023). Therefore, a process-based understanding of beach development of sheltered beaches is believed to be the way forward, such that the balance between local geometry, sediment type and hydrodynamic forcing can be resolved explicitly.

Detailed hydrodynamic studies of sheltered beaches are scarce (Vila-Concejo et al., 2010), but are needed as it is not evident that the skill of engineering morphodynamic models, which have been tried-and-tested on exposed coasts (e.g., Huisman et al., 2019; Luijendijk et al., 2017; Hegermiller et al., 2022; Splinter and Palmsten, 2012), directly translate to low-energy environments too. This is specifically relevant for the following aspects:

First of all, in low-energy environments, tidal and wind-driven currents are expected to play a relatively larger role in
sediment mobility and profile shape compared to beaches in wave-dominated environments (Bernabeu et al., 2003; Jackson and Nordstrom, 1992; Nordstrom and Jackson, 2012; Colosimo et al., 2023). For example, tidal currents enhance wave-induced bed-shear stresses (Kleinhans and Grasmeijer, 2006) and may simultaneously alter the dominant transport direction from cross-shore to alongshore (Héquette et al., 2008).

Second, the sea state itself is expected to differ between sheltered and open coasts. At sheltered sites, the wave state is
generally locally generated. Wave spectra of young sea states are more isotropic than their deep-water counterparts and are therefore expected to have a wider directional spreading as well as a a more gradual decay of energy with frequency (Donelan et al., 1985; Young et al., 1996; Ewans, 1998) compared to fully developed sea states. Additionally, the wave direction is modulated by the tide through the interaction of waves with the tidal current (Hopkins et al., 2016). These processes affect the strength and direction of wave radiation stresses, and thus the extent to which the wave field drives alongshore currents that
could transport sediment (Feddersen, 2004).

Third, the restorative capacity of wave-driven onshore transport by returning sediment high in the beach profile from lower parts after storm erosion (Hoefel and Elgar (2003)) is limited on low-energy beaches (Hegge et al., 1996; Jackson et al., 2002; Nordstrom and Jackson, 2012). The active part of the profile at a given moment, i.e., the part of the cross-section in which sediment is moved up and down by waves, is smaller on low-energy beaches (Hallermeier, 1980; Ton et al., 2021), meaning
that the tide will play a larger role in vertically displacing the sediment (Valiente et al., 2019), corresponding to larger relative tidal ranges (Masselink and Short, 1993).

Some sheltered coastlines have been reinforced with nourishments in anticipation of sea level rise (Perk et al., 2019; Ton et al., 2021). Although the use of sand has been postulated to be a sustainable approach to coastal maintenance of wave-dominated coasts (Stive et al., 2013; Grunnet et al., 2005; Brand et al., 2022), sandy reinforcements in low-energy environments
(Jackson et al., 2002; Vila-Concejo et al., 2010; Ton, 2023) are not as established. This partly explains why well-documented field observations of nourishment evolution at these sheltered sites are very scarce.

At these sites, the nourished sediment composition does not always match the natural gradation. Sediment heterogeneity, although widely recognized as an important control in beach development (e.g., Huisman et al., 2016; Bergillos et al., 2018), is generally inadequately accounted for or resolved in sediment transport models. Hiding and exposure of grains with widely varying diameters due to the way they are vertically sorted significantly complicates the response of the bed to hydrodynamic forcing (Kleinhans, 2005; McCarron et al., 2019), as the entrainment threshold of the different grains can be highly modified (e.g., Choule J. Sonu, 1972; Guillén and Hoekstra, 1996; Masselink et al., 2010; Richmond and Sallenger, 1985). Especially on low-energy beaches, which are often supply-limited, mixed sediments with a distinctive coarse fraction forming an armor/wear layer (Strypsteen et al., 2021) are an appealing option to limit erosion. Yet, how the local grain-size distribution and its spatial heterogeneity resulting from the implemented sediment mixture affects sediment pathways at a mixed-energy site requires more field data before transport models can be improved accordingly.

The extent to which these aforementioned processes are captured in engineering-type models requires validation. Therefore, there is a need for detailed intra-wave information on the vertical velocity profile, as well as the grainsize sorting and wave shape on top of bulk statistics such as mean water levels, and significant wave heights and periods at sheltered coasts. Here, we capitalize on the realization of a sandy nourishment bordering the Dutch Wadden Sea to provide such data. We collected high-resolution (in space and time) data on hydrodynamic forcing, bed composition and bathymetric changes. This paper presents data of a 6 week campaign (referred to as SEDMEX) as well as background data of bed levels and sediment composition data in the 2 years prior.

The overall aim of this study is to map the forcing mechanisms of sediment transport in a mixed-sediment, low-energy system. These data can be used to unravel generic processes (e.g. on sediment sorting and wave non-linearity) that are present at many sheltered coastlines worldwide. Moreover, the data enable the validation of model parameterization of unresolved processes (e.g., wave non-linearity, wave breaking, multi-fraction sand transport) in engineering-type models at sheltered beaches. This dataset is made available for use by the broader coastal research and engineering community as a benchmark test case for process-based models of sheltered systems with complex sediment mixtures.

## 2 Field site Prins Hendrik Zanddijk, the Netherlands

The campaign was undertaken at the Prins Hendrik Zanddijk (PHZD), a sandy coastal defense constructed in 2019 along the Wadden Sea coast of the island Texel (Figure 1a). This part of the coastline was originally protected by an asphalt dike (Figure 1b), but in light of anticipated sea level rise did not meet the Dutch safety standards any longer. Instead of heightening and widening the dike itself, a sandy foreshore including a sub-areal dune was constructed in front of the dike on the subtidal shoal Schanserwaard (Perk et al., 2019). The sandy foreshore was designed wide and high enough to take over the safety function of the dike entirely. The sandy foreshore consists of an 8 m high dune parallel to the dike and an attached 3 m high sand spit that created a shallow lagoon behind the dike and the spit, enabling the seaward discharge from an existing water pumping station meanwhile providing habitat for foraging birds (Figure 1b and Perk et al. (2019)). The composition of the bed of the Schanserwaard is mixed: both sand and silt are found. The core of the nourishment was constructed with fine sand

($d_{50}$ 200 $\mu$m), similar to the native sand. A 0.5-1 m thick top-layer of coarser sand ($d_{50} > 400$ $\mu$m) along the water line reduces sediment mobility. The sub-aerial section of the spit is additionally armoured with a top layer of shells to reduce aeolian transport (Perk et al., 2019; Strypsteen et al., 2021).

       The SEDMEX campaign was carried out 2 years after completion of construction. Therefore, it does not incorporate the initial profile adjustments of the engineered beach face to the natural system, but it does include morphological adaptations as

a result of forcing imbalances that act on the 1-5 year time scale.

       The Wadden Sea near Texel is connected to open sea through the tidal inlet Marsdiep. The Marsdiep branches into several channels, of which the basin's main channel Texelstroom (up to 25 meter depth) flows parallel to the field site. Tides in the basin are semi-diurnal with MHW=NAP+0.6 m and MLW=NAP-0.7 m. NAP is the local Dutch datum, roughly equal to mean sea level. The vertical tidal range varies from 1 to 2 m between neap and spring tide. Tidal velocities in the Texelstroom along

the PHZD reach up to 1 m/s and can increase further under influence of local winds. On the subtidal shoal, current magnitude is strongly reduced by bottom friction and were observed up to 40 cm/s. The majority of waves approaching the PHZD here are short-crested ($T < 5$ s), locally generated within the basin over a fetch of 5-25 km depending on the wind direction. However, under specific conditions, swell waves from the SSW propagate into the basin through the Marsdiep and partially refract towards the coastline too, but with diminishing amplitude. These swell waves are not deemed an important cause for beach

erosion by themselves, but can add to the overall impact when superimposing the locally generated wind sea.

       Measurements in this campaign were collected near the water line at the seaward side of the nourishment.

## 3   SEDMEX campaign

The field campaign was undertaken in autumn 2021 between September-09 and October-19. The measurements consisted of an array of instruments to capture the cross-shore wave and current transformation, and an alongshore array of wave and current

observations to investigate alongshore variability in hydrodynamics. Wave height and periods were recorded from pressure recording instruments. Directional wave parameters and near bed orbital and mean velocities were computed from velocity recording instruments. For their locations see Figure 2. Starting September-25, these wave observations were augmented with wave observations in the tidal channel Texelstroom (at 20 m water depth). Here, a Xylem Motus heave-pitch-roll wave buoy was deployed, measuring at 4 Hz.

On the spit beach, the wave transformation over the subtidal flat towards the beach was monitored over a cross-shore array. Along the nourishment, the alongshore variability was monitored with a series of instruments at constant water depth.

       The instrument set-up of the cross-shore array (Figure 3), focuses on high-frequency observations of near-bed orbital velocity, pressure and currents to be able to study intra-wave processes. All instruments were mounted to poles jetted into the bed and/or frames attached to the poles. The array consisted of 7 Acoustic Doppler Velocimeters (ADV's). Of those, 4 were Nortek

Vectors, sampling velocity at 16 Hz, 15 cm above the bed. Three of these ADV's were installed with additional RBRsolo[3] pressure sensor measuring at 8 Hz, also positioned 15 cm above the bed to obtain pressure measurements at equal height above the bed as velocity measurements. Three vertically stacked Sontek ADV's were positioned centrally in the array at NAP-0.85

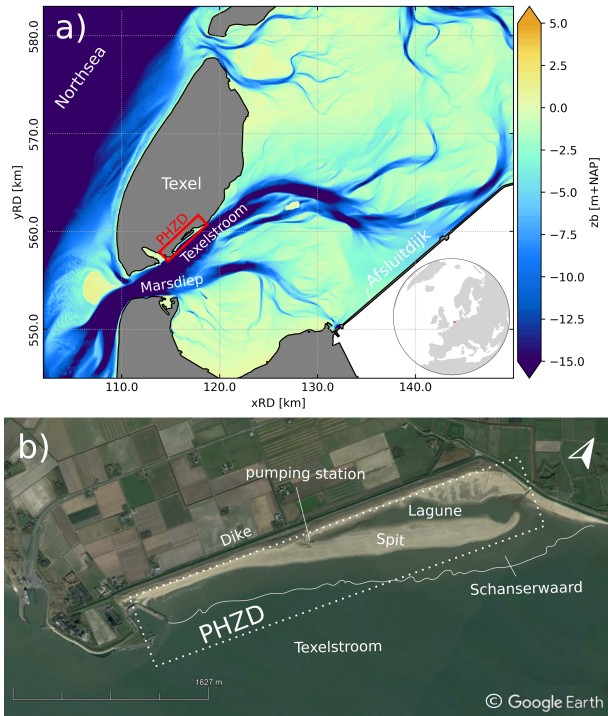

**Figure 1.** a) Location of the PHZD on the south-east side of the Dutch island Texel along the Wadden Sea. Colors indicate the bed elevation in m NAP with NAP being the local datum at approximately mean sea level, b) Aerial image of the field site. Sandy parts visible in the dotted rectangle are the sub-aerial parts of the PHZD nourishment.

m, measuring at 10 Hz. Three Ocean Sensor Systems OSSI pressure sensors spaced equidistantly between the most offshore ADV and the nearshore array were deployed at 15 cm above the bed measuring at 10 Hz. A downward looking Nortek Acous-

tic Doppler Current Profiler (ADCP) was placed in front of the beach face at NAP-1.2 m, measuring at 4 Hz in 50 mm bins. Sampling frequency between instruments varied because configuration possibilities differed between manufacturers. In general and when possible, we aimed to resolve wave processes with at least 10 Hz. The location on the cross-shore profile and the instrument's configuration are summarized in Table 1. Throughout the experiment, bed levels at the instrument locations varied for instruments close to the water line. The instrument heights were adjusted several times throughout the campaign to match

the target height above bed after morphodynamic changes. The horizontal x,y-position of the instruments was not changed.

Alongshore, 6 Nortek Vector ADV's at NAP-0.75 m and 5 OSSI pressure sensors at NAP-1.45 m were deployed that allow for studying alongshore variability in presence of North Sea swell, residual flow and local windsea. Figure 2 shows their position along the PHZD. Similar to the instruments in the cross-shore array, the instrument heights were adjusted throughout the campaign to match the target height above bed.

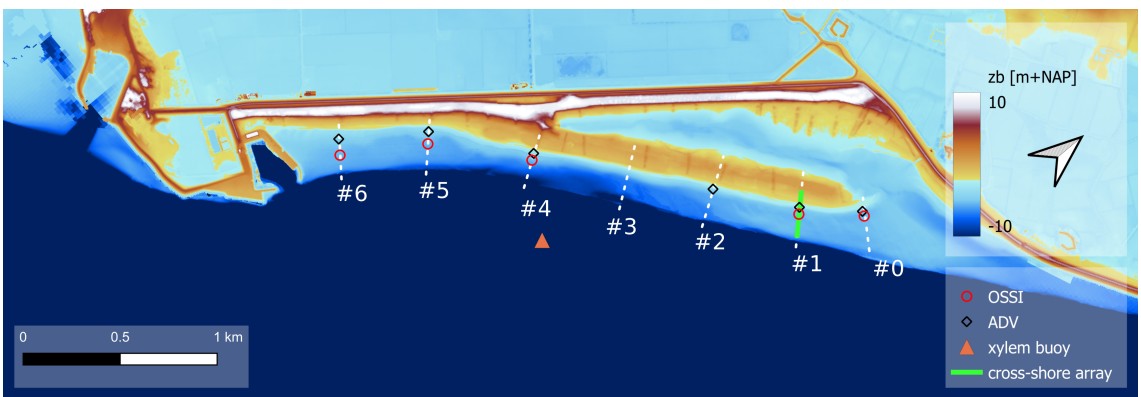

**Figure 2.** Nearshore bathymetry and topography. Markers indicate location of alongshore instruments during SEDMEX campaign and location of the cross-shore array is indicated by the solid-green line . Reference position of monitored cross-shore transects are shown in dashed white lines labelled by transect number.

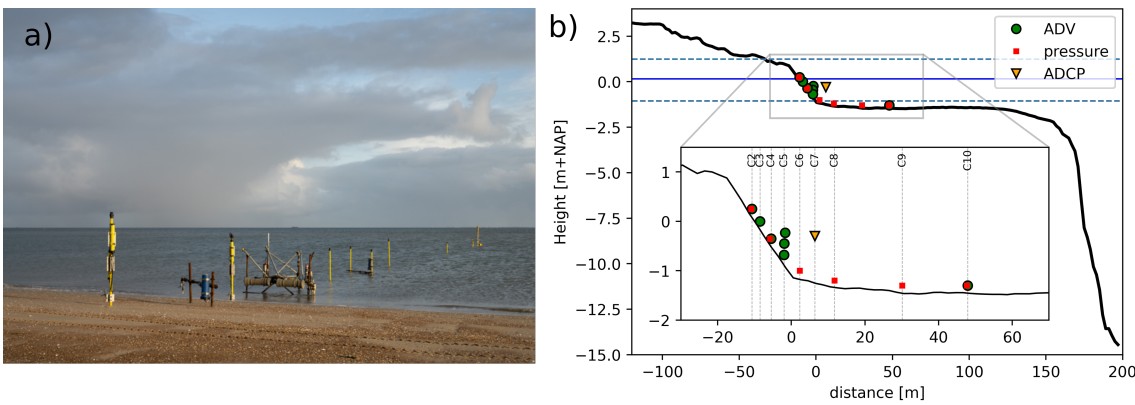

**Figure 3.** a) Impression of the cross-shore instruments and their mounts. b) Location of the instruments on the cross-shore array. Bathymetry at the beginning of the campaign in solid black lines, mean recorded water level in solid blue and maximum and minimum recorded water levels during SEDMEX campaign in dashed blue.

A series of Seapoint Turbidity Meters (STMs) and Campbell OBS-3+ turbidity meters were co-located and synchronised with a selection of the current meters (ADVs and ADCP) to collect proxy data on suspended sediment concentration at 15 cm above the bed (Table 1).

## 4 Beach profile measurements

At the position of the instruments, beach profiles were collected on a a bi-daily basis during low water with RTK-GNSS on top
of a walking wheel to obtain continuous surveys from the dune, throughout the surf zone to approximately 1 m water depth.

**Table 1.** Configurations of the deployed hydrodynamic instruments

| Instrument | Brand | Sampled quantity | Sampling frequency (Hz) | Height above bed (cm) | Initial bed level [m+NAP] | Configuration | Recording setting |
|---|---|---|---|---|---|---|---|
| L2C2SOLO | RBR Solo[3] | pressure | 8 | 15 | 0.1 | | continuous |
| L2C2VEC | Nortek Vector | velocity | 16 | 15 | 0.1 | sideways looking | 30min bursts 10s I.O. |
| L2C3VEC | Nortek Vector | velocity | 16 | 15 | -0.15 | downward looking | 30min bursts 10s I.O. |
| L2C4SOLO | RBR Solo[3] | pressure | 8 | 15 | -0.5 | | continuous |
| L2C4VEC | | velocity | 16 | 15 | -0.5 | sideways looking | 30min bursts 10s I.O. |
| L2C5SONTEK3 | SONTEK | velocity | 10 | 65 | -0.88 | downward looking | 30min bursts 60s I.O. |
| L2C5SONTEK2 | SONTEK | velocity | 10 | 40 | -0.85 | downward looking | 30min bursts 60s I.O. |
| L2C5SONTEK1 | SONTEK | velocity | 10 | 15 | -0.83 | downward looking | 30min bursts 60s I.O. |
| L6C1VEC | Nortek Vector STM | velocity turbidity | 16 | 15 | -0.80 | downward looking | 30min bursts 10s I.O. |
| L5C1VEC | Nortek Vector STM | velocity turbidity | 16 | 15 | -0.80 | downward looking | 30min bursts 10s I.O. |
| L4C1VEC | Nortek Vector OBS-3+ | velocity turbidity | 16 | 15 | -0.85 | downward looking | 30min bursts 10s I.O. |
| L3C1VEC | Nortek Vector STM | velocity turbidity | 16 | 15 | -0.80 | downward looking | 30min bursts. 10s I.O. |
| L1C1VEC | Nortek Vector STM | velocity turbidity | 16 | 15 | -0.80 | downward looking | 30min bursts 10s I.O. |
| L4C1SOLO | RBR Solo[3] | pressure | 8 | 15 | -0.85 | | continuous |
| L2C6OSSI | OSSI | pressure | 10 | 15 | -1.15 | | continuous |
| L6C2OSSI | OSSI | pressure | 10 | 24 | -1.45 | | continuous |
| L5C2OSSI | OSSI | pressure | 10 | 24 | -1.45 | | continuous |
| L4C3OSSI | OSSI | pressure | 10 | 24 | -1.45 | | continuous |
| L1C2OSSI | OSSI | pressure | 10 | 24 | -1.5 | | continuous |
| L2C7ADCP | Nortek Aquadopp STM | velocity profile turbidity | 4 | 95 | -1.2 | downward looking | 30min bursts 10s I.O. |
| L2C8OSSI | OSSI | pressure | 10 | 15 | -1.35 | | continuous |
| L2C9OSSI | OSSI | pressure | 10 | 15 | -1.45 | | continuous |
| L2C10SOLO | RBR Solo[3] | pressure | 8 | 15 | -1.45 | | continuous |
| L2C10VEC | Nortek Vector | velocity | 16 | 15 | -1.45 | downward looking | 30min bursts 10s I.O. |

Each survey with the RTK-GNSS covered 7 target transects, see Figure 2. The bed level changes along these transects are part of a longer-term measurement series. All site visits spanning the period between 2019 and 2022 are available in the published dataset.

To project the 1 Hz walking-wheel measurements to the same grid, each sample was allocated to its closest target transect coordinate, where the target transect was discretized with 0.25 m resolution. In case a sample was distanced more than 20 m from the target transect, it was discarded. Then, the bed level at the target coordinates was set to the average of all allocated GNSS-samples in the 0.25 m cross-shore section. In case no samples were assigned to the target coordinate, the bed level was set to NaN (Not a Number). NaNs are present in the dataset for some dates and tracks at deeper coordinates, when set-up of the water level prohibited us from going any deeper into the water with the GNSS device. Additionally, there are a few NaNs in the dataset due to occasionally walking faster than the 0.25 m cross-shore resolution divided by the sampling frequency.

## 5   Sediment sampling

Statistics on surface sediment samples together with RTK-GNSS location data are available for the PHZD intertidal beach. This includes compositional data of sediment samples collected along multiple cross-shore transects across the intertidal beach during the entire monitoring period (Figure 4a), and samples taken repeatedly at the same locations throughout multiple tidal cycles along one cross-shore transect (L2) during the SEDMEX period (Figure 4b). Additionally, a series of individual samples were taken at specific locations of interest such as the distal end of the spit.

Collection proceeded by inserting a small (6 cm high) empty jar upside down into the sand while rotating horizontally in an alternately (counter-)clockwise fashion (Figure 5a). Once filled up to a point where there was only about 1 cm left unfilled at the bottom of the jar, the jar was carefully turned upright whilst pulled out of the sand. An empty hand was used to quickly close off the opening to prevent any material from falling out in the process. The collected material was then siphoned into a pre-labeled plastic bag. However, on two days (at spring tide and neap tide), a sand-scraper device was alternatively used instead of a jar to obtain thin-layer samples (mm-scale) from the top 5 cm of the bed, which is in accordance with the sampling depth when using the jar. For more specifications of this device and details of the associated procedure the reader is referred to Van IJzendoorn et al. (2023). In all cases the filled sample bags were subsequently taken to a laboratory where their contents were oven-dried, mechanically sieved and weighed.

Drying of the sediment samples was achieved in one of two ways: (1) sub-samples of ∼150 g were collected inside aluminium cups, after which they were dried mostly overnight at a temperature of 105° C, or (2) the opened plastic sample bags were put in the oven in their entirety at a lower temperature of 65° C for a duration of at least 4 days (Figure 5b-d). Once completely dry, ∼150 g sub-samples were then emptied on top of a sieve tower standing on the platform of a mechanical shaker (Figure 5e).

The sieve shaker was turned on for approximately 15 minutes, after which the sieve tower was carefully removed from the platform. Each sieve was then weighed individually on an accurate semi-analytical balance. Measured weights were auto-

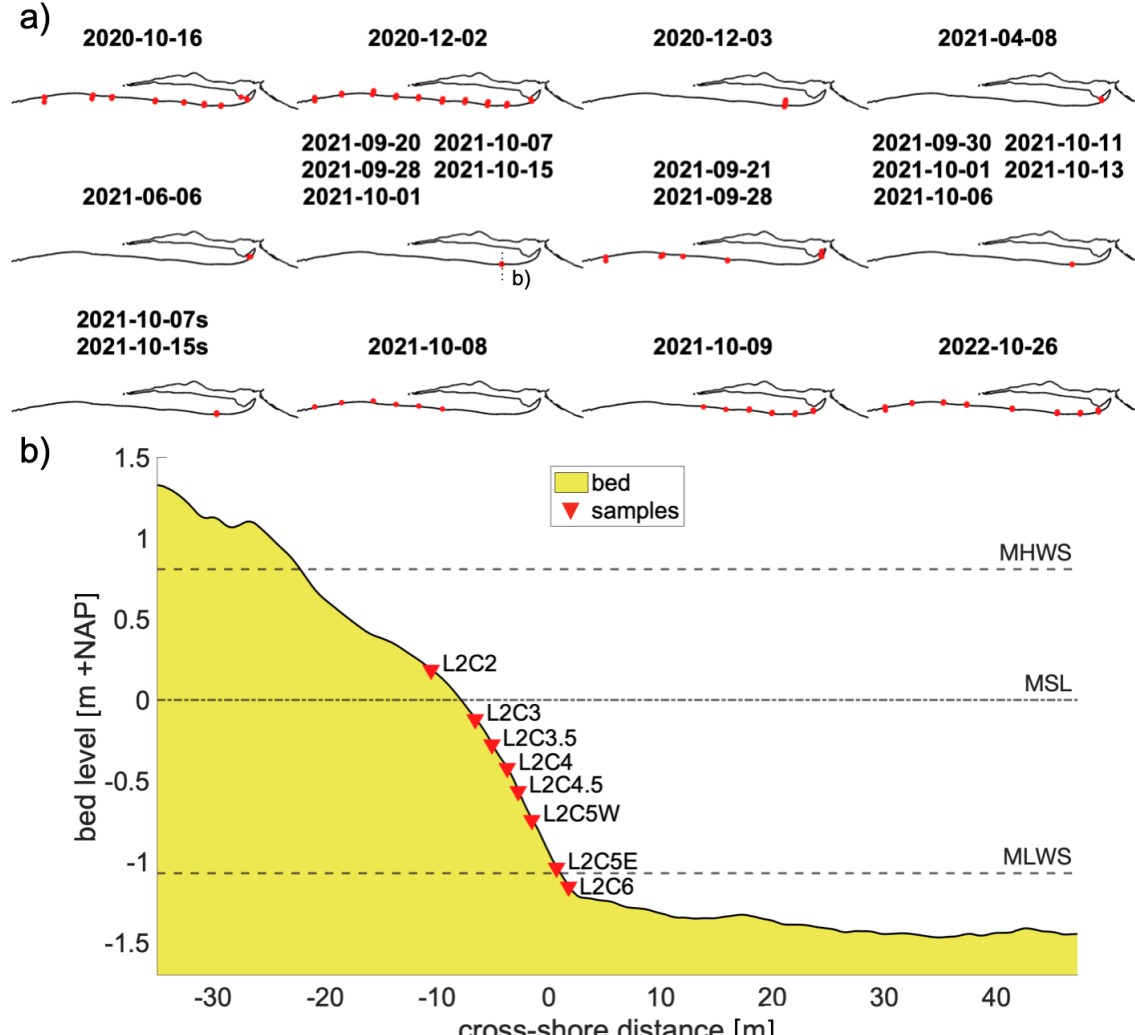

**Figure 4.** a) Overview of the horizontal locations of all sediment samples taken during the monitoring period. Multiple dates per subplot indicates samples were repeatedly taken on the exact same locations. On the two dates ending in an 's' the sand-scraper method was used. b) Vertical positions of the periodically collected samples along the L2 cross-shore instrument array. Horizontal dashed lines indicate the mean high water spring level (MHWS), mean sea level (MSL) and mean low water spring level (MLWS), respectively.

matically inserted into a spreadsheet next to their previously measured empty weights. Snapshots of the entire procedure are presented in Figure 5.

Shells or their fragments were not removed from the mixture beforehand, so the analysed samples are composed of both biogenic and non-biogenic minerals. The reason for doing so was to capture representative in-situ samples of the sediment that is transported by waves and currents. The generally flatter-shaped shell particles tend to end up in a sieve with an aperture close to the size of their second longest dimension, which is often substantially longer than their nominal length (i.e., as if they were

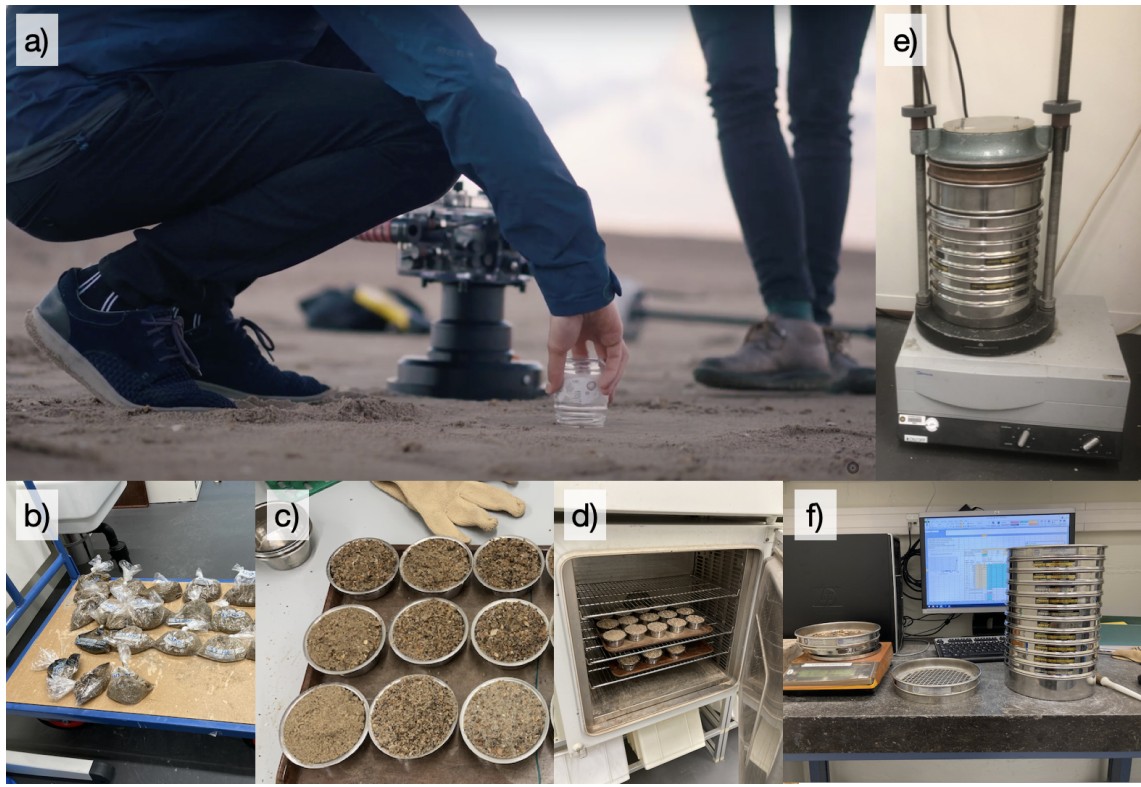

**Figure 5.** Sediment-analysis procedure: (a) sample with a small glass jar, (b) collect material in plastic bag, (c) collect sub-sample inside aluminium cup, (d) dry sub-samples in oven, (e) shake sub-sample in sieve tower and (f) weigh each sieve with retained material.

spherical). As the mineral density of many calcium-carbonate shell types is similar to that of quartz sand (i.e., ∼2650 g/L), this
could result in an inaccurate size-by-weight distribution of the whole sample. However, in our samples, the non-biogenic sand
fractions were much more abundant and, moreover, the biogenic fractions consisted predominantly of broken shell fragments,
with substantially smaller length to thickness ratios than intact shells. Therefore, it is assumed that the effect of the biogenic
material on the sediment-sample statistics is limited.

Mass data on the size fractions of the sediment samples were further processed using the open-access software GRADISTAT
(Blott and Pye, 2001), to obtain grain-size distributions as well as a commonly reported statistics such as grain-size percentiles
and standard deviations. All samples collected spanning the period between 2019 and 2022 are made available in the published
dataset. In addition to the cumulative percentile values, the following statistics on the grain-size distributions are presented:
mean ($M_G$), sorting ($\sigma_G$), skewness ($Sk_G$) and kurtosis ($K_G$). These are computed following Blott and Pye (2001) based on
the geometric (modified) graphical measures of Folk and Ward (1957).

## 6 Hydrodynamic data processing

The presented hydrodynamic dataset consists of three folders: (1) raw data as recorded by the instruments, converted to a standardised data format in netCDF including all meta-data required for further interpretation, (2) quality checked (QC) data and (3) tailored data, with burst-averaged wave and current characteristics.

For all instruments apart from the ADCP, it was decided to perform analysis on 10 minute data blocks instead of the full 30 minute bursts. At this site with short crested waves, 10 minute blocks are sufficiently long for mean spectral statistics in the sea-swell band and 10 minute blocks increase the total duration of valid measurement blocks with submerged instruments. Therefore, the raw data folder consists of data structured in 30 minute bursts, and qc and tailored data folders contain data structured into 10 minute bursts.

For each block, outliers were identified and samples deemed too noisy were masked with a separate protocol for each instrument type as further discussed below. In case the signal contained sufficient valid samples ($< 5\%$ masked), the masked samples were replaced with cubic interpolation of the time-series, presented as a time series of statistics on a 10 (30 for ADCP) minute resolution. All published data is recorded and saved in Central European Summer Time (GMT+2). The remainder of this section outlines the undertaken processing steps for each instrument group on top of the process described above.

### 6.1 Pressure

Pressure was recorded at the stand-alone pressure sensors, as well as the ADV's and ADCP. The recorded total pressure signals were corrected for air pressure fluctuations recorded on the dry beach. In the QC dataset, moments in time were the instrument fell dry were removed.

Then, near-bed pressure was converted to sea-surface elevation using linear wave theory:

$$P_{ss} = S_w^2 P_{hydrostatic} \tag{1}$$

$$S_w = \frac{\cosh(kd)}{\cosh(kh)} \tag{2}$$

with $P_{ss}$ the variance density spectrum of the sea surface, $P_{hydrostatic}$ the variance density of the hydrostatic surface elevation, $k$ the wave number, $d$ the water depth and $h$ instrument height above the bed. Due to the short-crestedness of observed waves, the sea-surface reconstruction was performed over the frequency band [0.05-1] Hz. This band deviates from common choices such as [0.05, 0.33] Hz (e.g. Neumeier and Amos (2006)). It was chosen such that sea-surface spectra for short-crested wind sea waves (with peak frequencies up to 0.5 Hz) are represented in the dataset. To prevent blow-up of noise in the power spectrum within the integration range, the linear transfer function, $S_w$ was capped to a maximum of $S_w = \max(S_w, 5)$. For example, for the deepest pressure sensor L2C10SOLO, on average 51% of the hydrostatic power density is present between [0.33, 1] Hz. This portion of sea surface variance density is on average amplified by a factor 260 in the linear reconstruction without a cap on $S_w$, but with the cap gets amplified by a factor 6. Using a cap in the sea surface reconstruction, the wave statistics are rather insensitive to the exact choice of the upper integration limit. For example, setting the upper integration limit to 0.8 Hz instead,

reduces the significant wave height on average by 1% and increases the mean wave period $T_{m01}$ by 1.2%. Some sensitivity to the choice of the cap value on transfer function $S_w$ however remains.

The cap on $S_w$ mitigates the blow-up of noise, but does not solve the problem that waves at high $kd$ are strongly attenuated at 15 cm above the bed. This results in unrealistically large $S_w$ values, that are replaced by the cap value in our approach. The lowest frequency on which the cap operates determines the reconstruction and is therefore a measure of the reliability of the linear reconstruction. When interested in statistics on spectral shape, the user could discard those observations where the onset of the cap on $S_w$ lies too close to the peak frequency $f_p$. At L2C10SOLO, the cap on $S_w$ onsets below $2f_p$ for 38% of observations.

The effect of currents on the sea-surface reconstruction is not incorporated in the linear reconstruction. This was decided as the effect of currents for those sensors where both flow and pressure observations were present was seen to be small, and moreover not all pressure sensors were accompanied by a flow sensor. At the deepest ADV L2C10VEC, the effect of the current on the relative wavelength was computed with the method of Guo (2002). For this, block-averaged horizontal velocities were decomposed in a component in direction of wave propagation and one perpendicular to that. On average, the root mean square error in wave length when neglecting currents was RMSE=0.3 m (3%).

Lastly, applying a linear sea surface reconstruction occurs under the assumption of a random-phase sea state, thereby ignoring the effects of non-linear processes (e.g. triad- and quadruplet wave-wave interactions) that phase couple sea-swell frequencies to higher harmonics that do not adhere to the linear transfer function (e.g. Martins et al., 2021). This means that potentially, bound wave variance, if present, can get over represented in the reconstructed sea-surface signal. The user should be aware of this if higher-order statistics of the reconstructed sea surface (not part of the published tailored dataset) are computed from the raw data.

The tailored dataset includes the following statistics: Significant wave height $H_{m0} = 4\sqrt{m_0}$, a few mean wave periods: $T_{m-1,0} = \frac{m_{-1}}{m_0}$, $T_{m0,1} = \frac{m_0}{m_1}$, $T_{m0,2} = \sqrt{\frac{m_0}{m_2}}$ with $m_n$ is the spectral moment of order $n$, the wave peak period and the smoothed wave peak period $T_{ps}$. $T_{ps}$ is the maximum of a parabolic fitting through the highest bin and two bins on either side the highest one of the discrete wave spectrum. The wave number $k$ is included on the tailored dataset too. If required, the user is able to correct wave numbers for ambient currents at the ADV sensors from the QC dataset themselves.

Furthermore, wave non-linearity parameters skewness and asymmetry were included and computed from the near-bed pressure as:

$$Sk = \frac{<p^3>}{<p^2>^{\frac{3}{2}}} \tag{3}$$

$$As = \frac{<\mathcal{H}\{p\}^3>}{<p^2>^{\frac{3}{2}}} \tag{4}$$

with $p$ the near-bed pressure signal in the frequency band [0.05-1] Hz, $\mathcal{H}p$ the Hilbert transformed pressure and $<>$ representing averaging over the 10 min block.

## 6.2 ADV and ADCP

All ADVs and the ADCP recorded velocities in probe-coordinates XYZ, and the probe orientation of each instrument was measured in the field. In the QC dataset all velocity measurements were rotated to East-North-Up coordinates and samples were flagged as outlier based on 3 conditions: (1) too low instantaneous inter-beam correlation, (2) exceeding the instantaneous recordable velocity threshold (varying between 1.5-2.1 ms$^{-1}$ for horizontal velocities dependent on horizontal or vertical placement of the instrument probe and 0.6 ms$^{-1}$ for vertical velocities), (3) statistical outlier detection: if the recording was more than 3 standard deviations from a block's mean it was flagged as outlier. Additionally, blocks were masked where the probe was less than 10 cm submerged based on the block's mean water depth. The correlation threshold was initially based on the criterion proposed by Elgar et al. (2005):

$$c_{min} = (0.3 + 0.4 \cdot \sqrt{(s_f/25)}) \cdot 100 \tag{5}$$

with $s_f$ the instrument's sampling frequency. This criterion rejected several blocks of which the block-averaged wave characteristics were well in line with the time series. Therefore, the QC dataset is published with a lowered inter-beam correlation criterion of ($> 50\%$), to arrive at a more complete time series of block-averaged wave and flow characteristics in the tailored dataset. The tailored dataset further includes orbital velocities in frequency band [0.05-1] Hz, mean wave direction $\theta_{mean}$ computed through linear wave theory and the method of maximum entropy (Lygre and Krogstadt, 1986), the component of the velocity signal in direction of wave propagation $u_d$ and non-linearity parameters for the near-bed orbital velocity in the direction of wave propagation and pressure in the frequency band [0.05-1] Hz. The computation of non-linearity in the orbital velocities follows Equations 3-4 but with the orbital velocity in the direction of wave propagation $u_d$ instead of $p$. For the ADCP, these wave statistics were computed from the velocities recorded in the cell 15 cm above the bed and additionally, block-averaged velocity profiles and depth averaged velocities were recorded on the tailored dataset.

Should a user want to apply a stricter correlation criterion to study for example turbulence, they can use the QC dataset as starting point and proceed from there.

Calibration of the turbidity signal to in-situ collected sand samples was performed in the lab, but suspended sediment concentrations (SSC) were difficult to relate to local hydrodynamics, most likely due to presence of mud in the basin (Pearson et al., 2021) and bubbles under breaking waves (Puleo et al., 2006). Therefore, the published QC dataset presents the unconverted turbidity signal, and SSC is not presented in the tailored dataset.

## 6.3 Overview quality checked hydrodynamic data

Figure 6 shows an overview of valid data bursts for all hydrodynamic instruments in the presented dataset. Missing data is caused by various reasons. The first, planned cause of missing data is that some of the instruments were placed at positions that fell dry during part of the tidal cycle. Therefore, all instruments placed higher in the profile than NAP-0.80m delivered intermittent time series, with water levels governing the gaps. Moreover, several instruments were removed and re-installed twice during the campaign for a cleaning of the probe, a battery pack change and data retrieval. This leads to data gaps

throughout the night of September-20 and the night of October-04. The RBR pressure logger L2C2SOLO was not redeployed after October-05. The Sontek ADV's ran on a single battery pack, which ran out of capacity around October-12.

Between September-28 and October-07, the beach profile at the cross-shore array underwent large changes and several instruments got buried, after which they were excavated and reinstalled, and got buried again. This has consequences for the data availability of L2C5SONTEK1, L2C4VEC, L2C4SOLO, L2C3VEC, L2C2VEC and L2C2SOLO.

Two instruments malfunctioned because of damage to the mounting material. Vector L3C1 was noticed to be rotated in its clamp on October-10, only possible after impact by a hard (floating) object. Analyses of the recorded velocities and directions revealed this must have happened on October-03. Therefore, the data between October-03 and October-10 were removed from the dataset. Similarly, the ADCP mounting frame was used by ignorant bypassing boaters as a mooring spot on October-09, which pulled the jetpoles skewed. The mounting poles were set straight again on October-10, intermediate data was discarded.

Lastly, some of the correlation and amplitude dropouts are believed to be caused by barnacle growth on transducer surfaces and/or seaweed getting stuck between legs of the probe. We visited the beach every other day and removed both barnacles and seaweed when identified, but had no control over what passed by in the intermediate times. We have relied on the quality control routine to remove bursts that suffer from seaweed, barnacles, or disturbance of the signal by cleaning the probe surfaces.

# 7 Environmental conditions

Figure 7 shows the wind speed and direction, water level, as well as significant wave height, mean period and near-bed velocities as measured on the deepest ADV (L2C10VEC) for the complete SEDMEX measuring campaign. The campaign captured two spring-neap tidal cycles, which allows for investigation of interaction between tides, wind-driven currents and wave-driven processes. Wind data is available from the nearest KNMI station De Kooy (10 km southward). The mean tidal range during the campaign was 1.25 m and varied between 0.5-1.8 m. The water level anomaly, estimated with a Godin filter of two times 24 hours and 1 of 25 hours, was positive for most of the campaign. Negative anomalies only occurred when the wind direction was Southwest to Northwest. The recordings started during a fair weather period starting September-10 to September-18. The most energetic event occurred in the week September-30 to October-05, when strong winds caused ∼30 cm setup (Figure 7b) in the tidal basin and south-southwesterly waves penetrated through the tidal inlet, partially refracting over the platform edge to the beach. On October-01, strong winds led to flow reversal over the entire platform, with flood-directed currents prevailing for three high tides (Figure 7c). A second period of very calm conditions started October-07 and lasted up to October-12.

In the offshore Texelstroom channel, significant wave heights reached up to 1 m, with corresponding peak wave period between 4-5 s (not plotted). Because of the platform depth of approximately NAP-1.4 m, waves of these heights broke on the platform edge. ADV L2C10VEC at NAP-1.45 m showed a maximum wave height of 60 cm and peak wave periods of 3-3.5 s, indicating that not all wave energy in the channel reaches the beach (Figure 7d). Mean wave period on the platform generally relates with wind direction: onshore-directed winds generate wind waves in the basin with short-crested, short-period waves. With shore-parallel winds from the south-west, the peak period increases due to the longer fetch length of waves that penetrate

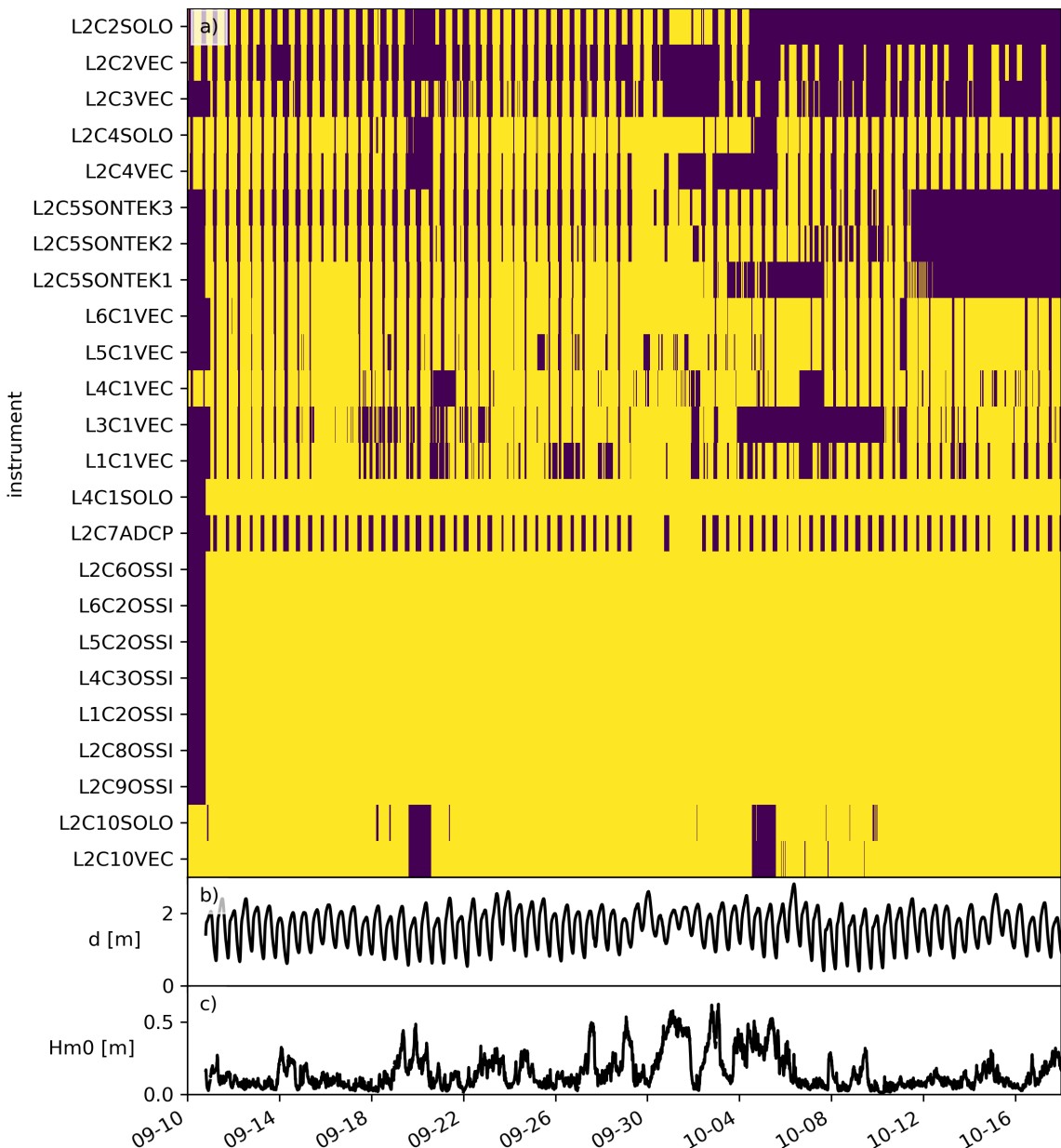

**Figure 6.** (a). Time series of available bursts of observations that passed the quality requirements, stacked per instrument. Yellow = available. Panels below show observations of (b) water level and (c) significant wave height at L2C9OSSI For reference.

the tidal inlet from the North Sea (Figure 7e). 10 Minute time-averaged near-bed velocities on the platform directly seaward of the beach face are predominantly alongshore directed (Figure 7c).

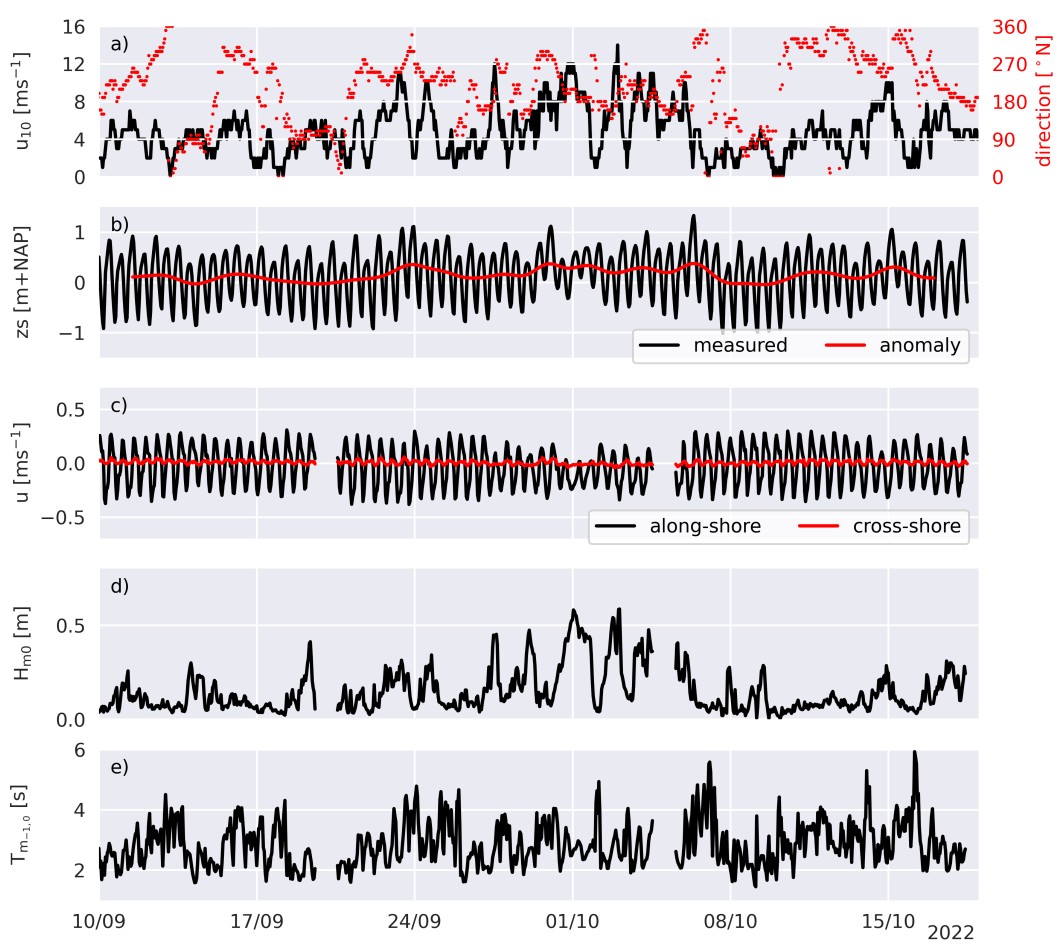

**Figure 7.** Hydrometeo conditions during the campaign: (a) wind speed and direction, (b) water level and water level anomaly, (c) alongshore and cross-shore flow velocity, (d) significant wave height at the deepest ADV on the platform and (e) mean wave period $T_{m-1,0}$ at that same ADV. The two data gaps are related to maintenance days.

## 8 Data samples

### 8.1 Cross-shore wave transformation

The cross-shore wave transformation governs where to expect wave-driven sediment transport and also determines whether this transport is onshore, offshore or alongshore directed. Figure 8 shows a snapshot of the recorded hydrodynamics on September-19 at 08:30 along the densely measured array. Positive onshore direction at this cross-section is 135 degrees counterclockwise from East, and positive alongshore direction is 225 degrees counterclockwise from East. Waves of 40 cm height propagated over the platform only to break close to the water line (Figure 8a). The waves were 20° oblique on the platform and turned

shore normal over the beach face (Figure 8c). Wave skewness develops over the last 20 m towards the water line (Figure 8d). Near-bed orbital velocities increases over the beach face (Figure 8e), and near-bed cross-shore velocities near the waterline are offshore directed. Alongshore (tidal) velocities first slightly decrease with water depth, but then the wave-driven alongshore component increases the total alongshore velocity again in even shallower water (Figure 8f).

### 8.2 Bed level changes

The cross-shore bed level was monitored throughout the SEDMEX campaign. During calm conditions with waves smaller than 30 cm, bed levels were stable. Several energetic days caused continuous profile adjustments. Those adjustments are captured in the dataset for six cross-sections along the entire beach with approximately two-day temporal resolution. Particularly in the period September-26 and October-07, bed level changes were observed along the spit (Figure 9). In this week, swash berms that had built up over the calm period in September were flattened out. On transect L0, approximately 3 $m^2s^{-1}$ was removed

from the profile above NAP+0.3 m. At this transect and on other transects along the spit (not shown here), this volume was found back lower in the profile. At the cross-shore array, a similar volume of 3 $m^2s^{-1}$ was added to the profile below NAP-0.3 m. Throughout this energetic period, the beach slope between MLW (NAP-0.7 m) and MHW (NAP+0.6 m) flattened from 1/6 to 1/10.

### 8.3 Grain-size distributions

Beachface surface samples at the PHZD that were collected in autumn of the years 2020, 2021 and 2022 showed an overall mean grain size ($M_G$) of 750 $\mu$m and on average a poor degree of sorting ($\sigma_G$ = 2.35). The poor sorting was often clearly visible from the often polymodal grain-size distributions. From an alongshore perspective, the $10^{th}$, $50^{th}$ and $90^{th}$ grain-size percentiles generally all followed a similar trend with the largest values found along the sand spit and lowest at the southern part of the beach. Within the more frequently sampled cross-shore array (Figure 4b), the mean of surface samples was coarser

($M_G$ = 1102 $\mu$m) but varied greatly between individual samples (often by several 100s $\mu$m) in both space, e.g., depending on the position of shoreline formations such as the beach step, beach cusps, swash bars and runnels, and in time, e.g., throughout a tidal cycle ($\sigma_G$ = 2.46). An example of a distribution is shown in Figure 10. Despite it being a relatively well-sorted example for this site ($\sigma_G$ = 2.14), the width of the distribution indicates the presence of a wide range of grain-size fractions.

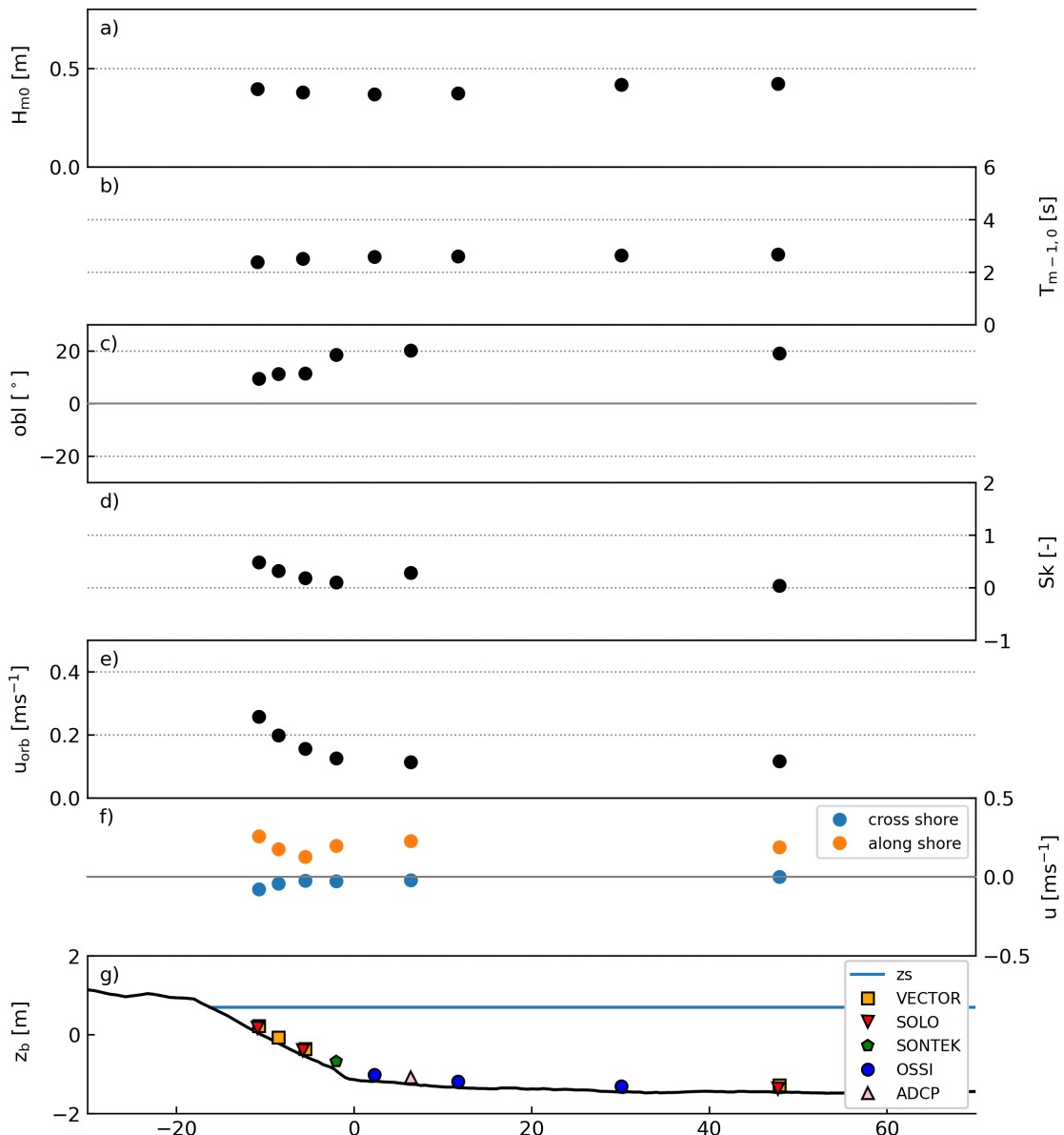

**Figure 8.** Snapshot of the observed cross-shore wave and current transformation along the cross-shore transect on September-19 08:30: a) significant wave height Hm0, b) mean wave period $T_{m-1,0}$, c) wave angle of incidence, d) near-bed velocity skewness, e) near-bed orbital velocity, f) burst mean currents in the cross-shore and alongshore direction (cross-shore positive is shoreward, alongshore positive is southwest ward), g) position of instruments on the profile with the burst mean water level for reference.

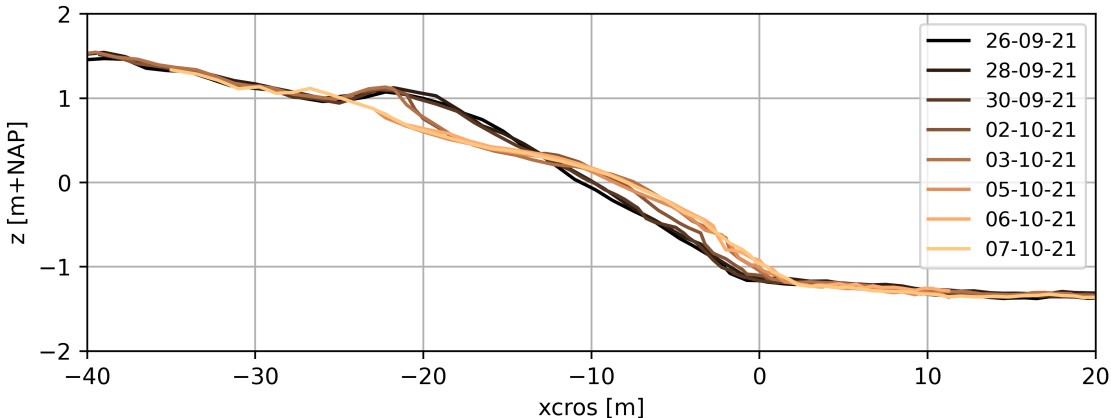

**Figure 9.** Cross-shore profile of the sand spit of the Prins Hendrik Zanddijk, measured before and after October-01 at L0 (See Figure 2 for location of L0 on the beach).

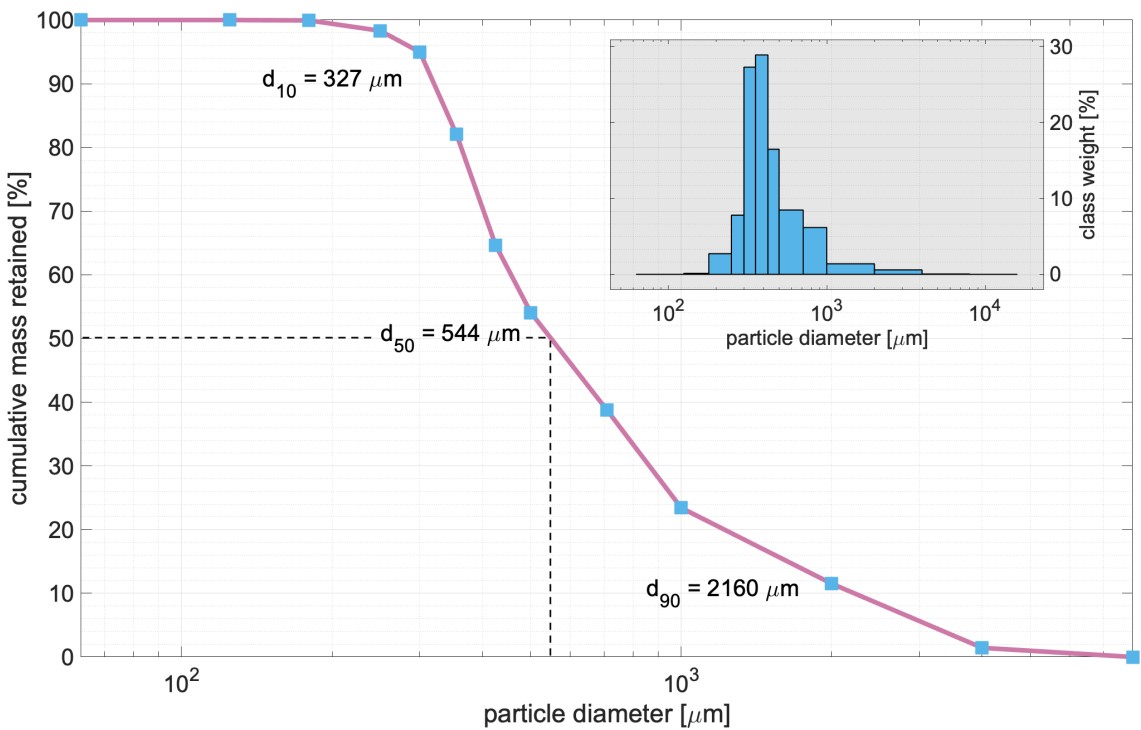

**Figure 10.** Example of a cumulative grain-size distribution curve and corresponding logarithmic frequency bar graph (embedded) of a sediment sample ('L2C3_1') taken at NAP-0.1 m on 7 October 2021. Black dashed lines indicate the median grain size of the sample. This sample was relatively well sorted for this site ($\sigma_G = 2.14$).

## 9    Conclusions

A comprehensive data set of sheltered-beach hydrodynamic forcing and beach response has been presented of the 6-week field campaign SEDMEX. This campaign was conducted on the Prins Hendrik Zanddijk, a sheltered man-made beach situated on the lee side of the barrier island Texel in the Marsdiep basin of the Dutch Wadden Sea. This data set allows in-depth analysis of the forcing mechanisms of sediment transport in this mixed-sediment, low-energy system, in particular the interplay between waves and currents and their effect on the beach face development. The duration of the campaign was sufficiently long to capture a wide range of environmental conditions, giving insight into the (relative) importance of bed-shear stresses induced by wind, waves and tides. Furthermore, sediment sorting processes can be studied based on the combination of sediment samples and detailed hydrodynamic measurements. Additionally, we envision this data set will support the validation of morphodynamic models (e.g. Delft3D, XBeach) on sheltered beaches. The high-temporal resolution observations additionally allow for testing of intra-wave scale models in complex real-world applications. Altogether, this comprehensive data set provides new data of a case study in the Wadden Sea (NL) to improve our predictive understanding of morphodynamics in sheltered (semi-enclosed) systems, where morphological evolution is governed by a subtle interplay between waves, tidal and wind-driven currents and bed composition due to low-energy (near-threshold) forcing.

## 10    Data availability

The data set presented in this article has been published at 4TU Centre for Research Data (van der Lugt et al., 2023) at DOI: 10.4121/19c5676c-9cea-49d0-b7a3-7c627e436541 following the FAIR principles (Wilkinson et al., 2016). The hydrodynamic data and profile development have been published in netCDF format and sediment statistics are published in CSV format. The underlying raw data as produced by the instruments together with the (matlab and python) scripts for conversion to netCDF with metadata are maintained under version control and are available upon request from the authors.

*Sample availability.*  Remaining dried sample materials are stored in dry storage at the Earth Simulation Laboratory, Princetonlaan 4, 3584 CB Utrecht, the Netherlands. All sample bags are labeled and in most cases contain less than 100 g of sediment, depending on the sample location on the beach (i.e., wetter generally means less sample). These quantities are generally too small for another mechanical-sieving procedure but can still be used for other analysis techniques which require much less material, for instance using a settling tube. Access to the sample collection is possible on request; please contact J.W. Bosma (j.w.bosma@uu.nl).

*Author contributions.*  MvdL: Methodology, Field work, Data Curation and Formal analysis hydrodynamics and beach profiles, Writing – original draft preparation, review and editing. JB: Methodology, Field work, Data Curation and Formal analysis sediment samples and turbidity, Writing – review and editing. MdS: Conceptualization, Supervision, Writing – review and editing. TP: Conceptualization, Supervision,

Writing – review and editing. MvM: Methodology, Field work. PvdG: Field work. GR: Conceptualization, Methodology, Supervision. AR: Methodology, Supervision, Writing – review and editing. SA: Conceptualization, Writing – review and editing.

*Competing interests.* All authors declare they have no competing interests.

*Acknowledgements.* This work is part of the research program: EURECCA 'Effective Upgrades and REtrofits for Coastal Climate Adaptation' under project number 18035, financed by NWO Domain Applied and Engineering Sciences and supported by Jan De Nul Group, Hoogheemraadschap Hollands Noorderkwartier, Waterproof B.V., Arcadis and Deltares. We thank the reviewers Arnaud Héquette and Kévin Martins for their valuable suggestions that really improved this manuscript. We thank dredging company Jan De Nul Group in particular for supplying the bathymetric data and deploying the wave buoy in the channel. We would like to thank the staff of the Utrecht University Earth
Simulation Lab, in particular Mark Eijkelboom, Henk Markies and Arjan van Eijk, and the staff of the TU Delft Fieldwork Lab, Chantal Willems, Arie van der Vlies and Arno Doorn for all efforts in preparation and execution of this campaign. Special thanks to the dedication of MSc students involved in the campaign: Jelle Woerdman, Martijn klein Obbink, Roel Hoegen, Ruurd Jaarsma and Menno van Maanen.

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
