# Peer review of "Measurements of morphodynamics of a sheltered beach along the Dutch Wadden Sea"

_Earth System Science Data, 2023_

## Referee Comment (RC2)

[Figure]

Figure 1: Reconstruction analysis for the calm conditions of 13 September 2021 at 12:45. The left panel shows the surface elevation energy density spectra $E$ for the two reconstructions (hydrostatic and linear). The central panels shows the sensitivity of the bulk parameters $H_{m0}$ and $T_{m02}$ to the choice of the high cutoff frequency in the integral for computing moments. The right panel shows the bicoherence of the hydrostatic reconstruction.

[Figure]

Figure 2: Reconstruction analysis for the calm conditions of 13 October 2021 at 00:15. The left panel shows the surface elevation energy density spectra $E$ for the two reconstructions (hydrostatic and linear). The central panels shows the sensitivity of the bulk parameters $H_{m0}$ and $T_{m02}$ to the choice of the high cutoff frequency in the integral for computing moments. The right panel shows the bicoherence of the hydrostatic reconstruction.

[Figure]

Figure 3: Reconstruction analysis for the energetic conditions of 19 September 2021 at 08:15 (conditions of Figure 8 in the paper). The left panel shows the surface elevation energy density spectra $E$ for the two reconstructions (hydrostatic and linear). The central panels shows the sensitivity of the bulk parameters $H_{m0}$ and $T_{m02}$ to the choice of the high cutoff frequency in the integral for computing moments. The right panel shows the bicoherence of the hydrostatic reconstruction.

[Figure]

Figure 4: Reconstruction analysis for the energetic conditions of 1 October 2021 at 02:45. The left panel shows the surface elevation energy density spectra $E$ for the two reconstructions (hydrostatic and linear). The central panels shows the sensitivity of the bulk parameters $H_{m0}$ and $T_{m02}$ to the choice of the high cutoff frequency in the integral for computing moments. The right panel shows the bicoherence of the hydrostatic reconstruction.

---

## Author Comment (AC1)

We thank Arnaud Héquette for reviewing our manuscript. The suggestions improved the dataset and accompanying manuscript. We address the specific comments one by one below. Original reviewer comments are shown in **black font**. Our responses and modifications affected in response to Reviewer comments are shown in **green font**. Modifications in the manuscript are shown in the **blue font**.

**Specific comments**

- line 13: The authors consider that "sheltered coastlines are traditionally defended by hard coastal structures made of concrete, asphalt or stones". This statement is a little surprising since sheltered coastlines are by definition protected from the action of high-energy waves that are responsible for coastal erosion. I would rather think that hard coastal structures would commonly be implemented along open coasts as a protection against high-energy waves.

We acknowledge that this wording was too strong and focussed on the field site we are studying. To make sure we address both this reviewer's and the other reviewer's concern about the introduction, we rewrote it slightly. In particular, we wrote an alternative opening paragraph of the manuscript:.

L13-16: Sheltered coastlines are protected from high wave impact and can be found in estuaries, coastal basins or inland lakes. Sandy sheltered coastlines typically undergo smaller rates of storm-driven erosion than exposed coasts and received less attention in research. Nevertheless, understanding the physics of these beaches is just as important, as they also protect vital coastal infrastructure and communities that rely on them as a defense against flooding

- lines 23-24: It is stated that "beaches in estuaries and on the lee-side of islands are not truly fetch-limited, as refracting ocean swell waves may still reach the shore…". Although it is true that some beaches in estuaries are not truly fetch-limited, other beaches located in an estuary may be fetch-limited where the mouth of the estuary is protected from ocean swells (by an island for example).

We acknowledge that our original wording here is too general. We have rephrased it to the following:

L20-21: Beaches in estuaries and on the lee-side of islands may be typically sheltered from long period swell, but under the right conditions some refracted ocean swell waves may still reach these shores (Cooper et al., 2007).

- lines 74-75: it is written that the data were collected "shortly after construction". After construction of what? Presumably, this refers to the sand nourishment that took place in 2019, but this is mentioned later in the paper.

Thanks for pointing this out. We have removed the reference to the construction of the sand dike here and explain about the timing of the campaign with respect to finalization of the construction in more detail in the next section:

L81-85:  The campaign was undertaken at the Prins Hendrik Zanddijk (PHZD), a sandy coastal defense constructed in 2019 along the Wadden Sea coast of the island Texel (Figure 1a). This part of the coastline was originally protected by an asphalt dike (Figure 1b), but in

light of anticipated sea level rise did not meet the Dutch safety standards any longer. Instead of heightening and widening the dike itself, a sandy foreshore including a sub-areal dune was constructed in front of the dike on the subtidal shoal Schanserwaard (Perk et al., 2019).

- lines 89-90: The authors indicate that "the campaign was undertaken at the Prins Hendrik Zanddijk (PHZD), a sandy coastal defense along the Wadden Sea coast of the island Texel constructed in 2019". Maybe some elements of context could be useful here. What motivated the realization of a beach nourishment on a sheltered beach? Was the coast eroding even if it is partially protected from offshore waves from the North Sea? Was the dyke damaged which required further coastal protection?

This was indeed very minimally described. We incorporated some more background information in the field site section. The resulting paragraph is the same that is included in the reply to the previous comment (L81-85).

- lines 93-97: Some information is given here about waves and tides at the study site, stating that most waves reaching the site are locally generated short waves, but that in certain conditions swell waves can propagate into the basin and can refract towards the coastline, but with diminishing amplitude. Given the configuration of the site, one would expect that wave heights would be strongly lower when reaching the beach and would not have enough energy for causing significant erosion. Is there any information on this?

It is also stated that tidal velocities can reach 1 m/s along the PHZD (or even more under the influence of local winds). Could tidal currents be a major cause of beach erosion at this site instead of waves?

We added two sentences to clarify a bit more on the role the swell waves are deemed to have and the magnitude of the tidal current on the platform the PHZD is constructed upon":

L100-101: On the subtidal shoal, current magnitude is strongly reduced by bottom friction and were observed up to 40 cm/s.

L104-105: These swell waves are not deemed an important cause for beach erosion by themselves, but can add to the overall impact when superimposing the locally generated wind sea

- lines 125-131 (and table 1): It is indicated that the different hydrodynamic instruments recorded data at different sampling frequencies, ranging from 4 to 16 Hz. Any reason why the instruments were programmed to make measurements at different sampling frequencies?

We targeted to measure wave related signals up or above 10Hz when possible, but instrument options varied. We worked with two type of acoustic doppler velocimeters (Nortek Vector and SONTEK ADV), 3 type of pressure sensors (Nortek Vector, OSSI wave gauges and RBR Solo's) and one type of acoustic doppler current profiler (Nortek ADCP). All of these instruments have their own range of possible sampling frequencies. The Solo's maximum sampling frequency was 8 Hz, the OSSI's could measure at either 2,5,10, 20 or 30 Hz and both vectors could measure at 2,4,8,16,32,64 Hz. Current profilers could measure up to 4 Hz. We addressed this in the manuscript as follows:

L126-127: Sampling frequency between instruments varied because configuration possibilities differed between manufacturers. In general and when possible, we aimed to resolve wave processes with at least 10 Hz.

- lines 179-180: the authors indicate that "shells or their fragments were not removed…, so the analysed samples are composed of both biogenic and non-biogenic minerals". What is the justification for not removing the shells or shell fragments? Although, some scientists prefer to carry out grain-size analysis on the total sediment samples without removing biogenic sediments (because the biogenic material is also a part of the sediment transported by waves and currents), some researchers prefer to remove the biogenic fraction from samples when measuring sediment grain-size. The authors mention that the non-biogenic sand fractions were much more abundant than the biogenic sediments, but it would have been better to carry out grain-size analyses on samples including both biogenic and non-biogenic sediments and on non-biogenic sand and provide the results of both analyses.

As mentioned by the reviewer, the choice to analyse the total samples of mixed sediments, without first separating the biogenic and non-biogenic components, was motivated by the goal of representing in-situ conditions as accurately as possible. Keeping shell and sand fractions combined in our sediment analysis allows us to examine the sediment sorting in relation to the morphological change in a holistic manner. We acknowledge that some researchers prefer to separate the biogenic and non-biogenic material for specific analyses. The densities of the most abundant types of shells (calcium carbonates) at our study site (e.g., of the species Cerastoderma edule and Spisula solida) are similar to that of quartz grains (i.e., ~2650 kg/m3), whereas their shapes are respectively flat (i.e., larger length to thickness ratio) and spherical. Although this discrepancy in particle properties may result in an overestimation of particle diameters from mechanical sieve analysis, we assumed this effect to be marginal in our case with underrepresentation (estimated typically <5 wt.%) of shell material (predominantly fragments, with small length to thickness ratios) in the bulk of most samples. Therefore, the added value of separating the carbonate from the siliciclastic material, by pre-treating the samples with hydrochloric acid, was not considered substantial for our study.

As a matter of fact, the current data set of sediment samples has made us question the role of shells and shell fragments, even in smaller quantities, in total sediment transport, incipient motion and sediment sorting by waves and currents.

L176-182: The reason for doing so was to capture representative in-situ samples of the sediment that is transported by waves and currents. The generally flatter-shaped shell particles tend to end up in a sieve with an aperture close to the size of their second longest dimension, which is often substantially longer than their nominal length (i.e., as if they were spherical). As the mineral density of many calcium-carbonate shell types is similar to that of quartz sand (i.e., ~2650 g/L), this could result in an inaccurate size-by-weight distribution of the whole sample. However, in our samples, the non-biogenic sand fractions were generally much more abundant and, moreover, the biogenic fractions consisted predominantly of broken shell fragments, which typically have substantially smaller length to thickness ratios than intact shells.

- lines 289-290: It is stated that "alongshore (tidal) velocities first decrease with water depth, but then the wave-driven alongshore component increases the total alongshore velocity again in even shallower water (Figure 8f)". Although Figure 8f actually shows an increase in alongshore current velocity with decreasing water depths on the beach face (from approximately 200 to 210 m), which is presumably related to a wave-driven alongshore component, the figure does not show a decrease in current velocity with water depth over the rest of the beach, but rather similar current velocities.

From both reviewer's feedback, we realized that there was a mistake in the computation of the significant wave height in the SOLO instruments as the sampling frequency was wrongly set to 10Hz instead of the correct 8 Hz in the processing. Correcting this has modified the wave height transformation of Figure 8. This makes the interpretation more logical, there is now no longer a wave height increase with decreasing water depth. The alongshore velocity in Figure 8 shows a small reduction from L2C10VEC through L2C5SONTEK1 to L2C4VEC (with decreasing water depth), after which it slightly increases again at L2C3VEC and L2C2VEC. We have updated Figure 8.

- Figure 2: It could be useful to add the number of the beach profile on each transect shown in the figure. This would be helpful for quickly locating the position of the profile along the beach when using the beach profile data.

Thanks for this suggestion. We have indeed included the profile numbers in the figure.

- Figure 8: The graph 8f shows mean current speeds in the cross shore and alongshore directions with positive and negative values. It could be useful to indicate if positive (negative) values of cross shore currents are onshore-directed (or offshore-directed); this could be mentioned in the figure caption or directly in the graph. The same for alongshore currents: do positive or negative values indicate eastward- or westward-flowing currents?

Thanks for this suggestion. We have modified the caption as follows:

Figure 8. Snapshot of the observed cross-shore wave and current transformation along the cross-shore transect on September-19 08:30: a) significant wave height Hm0, b) mean wave period Tm−1,0, c) wave angle of incidence, d) near-bed velocity skewness, e) near-bed orbital velocity, f) burst mean currents in the cross-shore and alongshore direction (cross-shore positive is shoreward, alongshore positive is south-west ward), g) position of instruments on the profile with the burst mean water level for reference.

---

## Author Comment (AC2)

We thank Kevin Martins for the detailed review and very thorough look at the dataset. The suggestions improved the dataset and accompanying manuscript. Below we provide our responses to the reviewer comments. Original reviewer comments are shown in **black font**. Our responses and modifications affected in response to reviewer comments are shown in **green font**. Modifications in the manuscript are shown in the **blue font**.

**General/Major comments:**

Overall, the manuscript is relatively well written and organised, but I found that the introduction could be improved. I feel that the authors fail to find the right balance between the specificity of the site, and how the dynamics at this location is representative and similar to other low-energy sites around the globe: in other words, is this dataset going to help find generic processes, which can in turn be useful elsewhere? Or instead, is this study more relevant to the North Sea context? There is a sort of review on "sheltered" beaches, whose definition is not completely clear (is this the role of this paper to define such morphological features anyway, I wonder?). Part of my feeling also comes from phrases jumping from the Dutch context to generalities, which are sometimes extrapolated. For instance, the first phrase: "Sheltered coastlines are traditionally defended by hard coastal structures made of concrete, asphalt or stones.". I am not a specialist but I do not think this actually the case everywhere.

We understand the reviewer's suggestions and questions on the introduction. We think the findings are applicable at many sheltered shorelines as the fetch limited conditions are often dominant at these beaches. We improved the introduction in which we removed some of the specificity of the site and explicitly mention that we believe that the dataset will lead to generic knowledge (e.g. on wave nonlinearly during these low energy, short period conditions). Publications on generic processes using this dataset are in the making. The revised introduction of the manuscript can be read in L13-79.

1 -The authors often refer to "sediment transport" as measured or, at least, that can be studied with the present dataset. But is this really the case? Turbidity measurements could not be transformed into SSC, so how can the dataset actually serve to validate sediment transport (not morphological changes)? Because of this, I would tone down the sediment transport part, and rather focus on a dataset capturing the morphological evolution of the site as well as describing the spatio-temporal evolution of sediment characteristics. It does not mean that the dataset is less useful to model the site morphodynamics, there just is less means to validate sediment transport directly.

The reviewer is right to point out that the initial ambition of the campaign, to measure both hydrodynamics and sediment concentrations, was not realized as we decided to not publish sediment concentrations derived from the optical backscatter data due to difficulties with flocs in the water column.

Apparently, the current formulation of the scope may leave the reader with the impression that sediment transport was measured. We take the reviewers comment at heart and adjusted the manuscript to make sure the scope of the dataset is clear to the reader. Therefore, we have reviewed all mentions of 'sediment transport' in the manuscript and made more specific references to the scope of the data that is published and presented. The list below shows all mentions of 'sediment transport' in the old manuscript and changes in the revised manuscript ( >> new text):

L8-10: The novelty of this data set lies in the detailed approach to resolve forcing conditions on a sheltered beach, where  morphological evolution is governed by a subtle interplay between tidal and wind-driven currents, waves and bed composition, primarily due to the low-energy (near-threshold) forcing.

L41-42:  (we removed this line)

L45-47: For example, tidal currents enhance wave-induced bed-shear stresses (Kleinhans and Grasmeijer, 2006) and may simultaneously alter the dominant transport direction from crossshore to alongshore (Héquette et al., 2008). (we left this sentence unchanged)

L51-52: These processes affect the strength and direction of wave radiation stresses, and thus the extent to which the wave field drives alongshore currents that could transport sediment (Feddersen, 2004). (we left this sentence unchanged)

L55-57: Third, the restorative capacity of wave-driven onshore transport by returning sediment high in the beach profile from lower parts after storm erosion (Hoefel and Elgar (2003)) is limited on low-energy beaches (Hegge et al., 1996; Jackson et al., 2002; Nordstrom and Jackson, 2012). (we left this sentence unchanged)

L61-63:  At these sites, the nourished sediment composition does not always match the natural gradation. Sediment heterogeneity, although widely recognized as an important control in beach development (e.g., Huisman et al., 2016; Bergillos et al., 2018), is generally inadequately accounted for or resolved in sediment transport models.

L68-70: Yet, how the local grain-size distribution and its spatial heterogeneity resulting from the implemented sediment mixture affects sediment pathways at a mixed-energy site requires more field data before transport models can be improved accordingly. (we left this sentence unchanged)

L71-72: The extent to which these aforementioned processes are  captured in  engineering-type models requires validation.

L83-85: The overall aim of this study is to unravel  the forcing mechanisms of sediment transport in this mixed-sediment, low-energy system. These data are usable for validating  model parameterization of unresolved processes (e.g., wave non-linearity, wave breaking, multi-fraction sand dispersal ) in engineering-type models at sheltered beaches.

L288-289: The cross-shore wave transformation governs where to expect wave-driven sediment transport and also determines whether this transport is onshore, offshore or alongshore directed. (we left this sentence unchanged)

L313-314: This data set allows in-depth analysis of the forcing mechanisms of sediment transport in this mixed-sediment, low-energy system.

L322-324: Altogether, this comprehensive data set provides new data of a case study in the Wadden Sea (NL) to improve our predictive understanding of morphodynamics in sheltered (semi-enclosed) systems, where  morphological evolution is governed by a subtle interplay between waves, tidal and wind-driven currents and bed composition due to low-energy (near-threshold) forcing.

2 – I have concerns over the processing of the bottom pressure, and the bulk wave parameters provided in the paper. My attention got caught especially in Figure 8, with the much larger Hm0 offshore compared to other sensors, without an obvious explanation. Reconstructing the free surface elevation from bottom pressure is a problem that I know well, and I am fully aware of the challenges in the present dataset for reconstructing a signal with such short waves, and sometimes in the presence of relatively strong non-linearities. In short, there is no ideal solution, but at least the problem should be acknowledged, and the related uncertainties into bulk parameters quantified. First, there is no mention of the cutoff frequency for the linear correction, except at line 209 where 1 Hz is noted but this is surely not the correct one. Here, I analysed several bursts of data at the OSSI L2C10 (similar behaviour is observed at C9, I have not checked at other sensors) as follows. 40-min of data were taken during 4 situations of "low" and "high" energy conditions (2 each), including the one chosen. Note that the behaviour I describe next could be found in most bursts I extracted and tested. Over this 4 bursts of data, I computed power spectra of the hydrostatic ($\zeta_{hyd}$ in figures)and linear ($\zeta_L$) reconstructions of the free surface elevation from the detrented pressure signal using 60 Hann-tappered blocks of 64 s overlapping by 50% (effective d.o.f. are ~112). From these, the significant wave height $H_{m0}$ and mean wave period $T_{m02}$ are computed using a range of different cutoff frequencies. Lastly, the bicoherence $b$, a measure of the bound energy at a given frequency, was computed from $\zeta_{hyd}$ following Hagihira et al. (2001). From the provided plots, several important remarks can be made:

- Bulk wave parameters are extremely sensitive to the cutoff frequency, especially $H_{m0}$. Let us consider the energetic example of Figure 8 (19 September 2021 around 08:15): integrating up to 0.5 Hz gives Hm0 ~ 0.5 m, while integrating up to 0.8 Hz gives more than the double! In terms of energy, that is a factor four. Here, it should be noted that by looking at the hydrostatic reconstruction, choosing 0.8 Hz as a cutoff does not look incorrect, since it still corresponds to energy levels 2 orders of magnitude above noise level.
- The cutoff frequency chosen by the authors seems to vary in time, and sometimes even between parameters. In order to retrieve the values stored in the 'tailored' data, a cutoff frequency at 0.6 Hz should generally be applied, but it actually seems to vary between 0.4 Hz (13 October 00:15) to 0.9 Hz (1 October 02:45), and potentially less or more, respectively, since I only checked a limited amount of data bursts.
- Non-linearities seem relatively strong, and interactions around the peak frequency or between distinct bands of frequencies seem both intense and pretty common in the present dataset. Those explain large fractions of bound energy at high frequencies, leading to a potentially large overestimation of the pressure-to-surface transfer function (e.g., Martins et al., 2021), which affects the computation of bulk parameters. As shown in some of the examples provided, the blow-up of the linear reconstruction occurs rather "fast" in some cases, e.g. well below frequencies where noise start to dominate the pressure signal.

As mentioned above, I acknowledge the fact that there currently does not exist a method adapted to such wave conditions (non-linear reconstructions being currently limited to weakly or moderately dispersive regimes). Furthermore, the raw pressure data is provided, so that informed users will be able to decide themselves how to compute bulk parameters. However, a large fraction if not most users will not be fully aware of the issues, and will be mostly interested in bulk products as computed by the authors. In this context, and in my opinion, the issues related to the reconstruction of such short waves often undergoing non-linear processes should be acknowledged, and the uncertainty on

bulk quantities shoud be estimated and discussed. Here I chose bursts of data at high tide, where we expect minor influence from the current, but considering the wave periods measured here, this is definitely another aspect that need to be discussed as it unsure whether the authors account for it for computing wavenumbers.

The reviewer points out an important aspect that we did not pay sufficient attention to in the original manuscript. We thank the reviewer for raising this point. Indeed, we did not explain in enough detail how the reconstruction of sea-surface variance from the near-bed pressure signal was performed. In fact, the Transfer Function $S_w = \cosh(kd)/\cosh(kh)$ was computed with linear wave theory but capped at an amplification factor of 5 for amplification of the power spectrum, to prevent blow up of noise below 1 Hz in the computed wave statistics. We have modified the manuscript accordingly.

As mentioned by the reviewer and his accompanying figures, choosing one appropriate cutoff frequency can be difficult in case of a linear correction of the pressure data. We use an amplitude cap on the correction (rather than a frequency cutoff) and with this cap, the wave statistics Hm0 and Tm02 are almost insensitive to the choice of the cutoff frequency if the cut-off frequency is chosen above the sea-swell frequencies. This can be seen in the provided plots in Figure 1 (right panels, green lines). Here, we produced figures similar to the ones of the reviewer, using 10-minute bursts of data (as this was the chosen burst length for this dataset) at the similar moments in time as the reviewer chose. We show the effect of the capped Transfer Function Sw on the reconstruction of the sea-surface variance density spectrum in comparison with an unbound application of the Transfer Function. We also show how wave statistics Hm0 and Tm02 then vary with the upper integration limit (cut-off) frequency. Because the hydrostatic variance at frequencies above 0.6 Hz is not negligible for the short-crested waves we recorded, we believe our method with a capped transfer function is preferable over a shorter integration limit up to e.g. 0.6Hz.

Then, the reviewer points out that linear theory has it limits and we should acknowledge those. To that end we added a paragraph on consequences of bound wave energy on the reconstruction. The lowest frequency on which the cap operates determines the reconstruction and is therefore a measure of the reliability of the linear reconstruction. At L2C10SOLO, the cap on Sw onsets below 2fp for 38% of observations, see the attached Figure 2.

Lastly, the reviewer asks for a discussion of the influence of currents on the published wave statistics. We made an analysis of the effect of currents at station L2C10VEC. At this station, we expect the effect of currents to be largest, as currents are predominantly alongshore and at this deepest ADV station the waves are still most obliquely incident. The attached Figure 3 shows a 2D histogram of the current-corrected and the uncorrected wave length. The statistics from this comparison were included in the text of the manuscript.

All the above was incorporated into the manuscript as follows:

L208-240: Then, near-bed pressure was converted to sea-surface elevation using linear wave theory:

$P_{ss} = S\_w \, P_{hydrostatic}$        (1)

$S_w = \cosh(kd)/\cosh(kh)$        (2)

with Pss the variance density spectrum of the sea surface, Phydrostatic the variance density of the hydrostatic surface elevation, k the wave number, h the instrument height above the bed and d the water depth. Due to the short-crestedness of observed waves, the sea-surface reconstruction was performed over the frequency band [0.05-1] Hz. This band deviates from common choices such as [0.05, 0.33] Hz (e.g. Neumeier and Amos, 2006). It was chosen such that sea-surface spectra for short-crested wind sea waves (with peak frequencies up to 0.5 Hz) are represented in the dataset. To prevent blow-up of noise in the power spectrum within the integration range, the linear transfer function, Sw was capped to a maximum of Sw = max(Sw, 5). For example, for the deepest pressure sensor L2C10SOLO, on average 51% of the hydrostatic power density is present between [0.33, 1] Hz. This portion of sea surface variance density is on average amplified by a factor 260 in the linear reconstruction without a cap on Sw, but with the cap gets amplified by a factor 6. Using a cap in the sea surface reconstruction, the wave statistics are rather insensitive to the exact choice of the upper integration limit. For example, setting the upper integration limit to 0.8 Hz instead, reduces the significant wave height on average by 1% and increases the mean wave period Tm01 by 1.2%. Some sensitivity to the choice of the cap value on transfer function Sw however remains.

The cap on Sw mitigates the blow-up of noise, but does not solve the problem that waves at high kd are strongly attenuated at 15 cm above the bed. This results in unrealistically large Sw values, that are replaced by the cap value in our approach. The lowest frequency on which the cap operates determines the reconstruction and is therefore a measure of the reliability of the linear reconstruction. When interested in statistics on spectral shape, the user could discard those observations where the onset of the cap on Sw lies too close to the peak frequency fp. At L2C10SOLO, the cap on Sw onsets below 2fp for 38% of observations.

The effect of currents on the sea-surface reconstruction is not incorporated in the linear reconstruction. This was decided as the effect of currents for those sensors where both flow and pressure observations were present was seen to be small, and moreover not all pressure sensors were accompanied by a flow sensor. At the deepest ADV L2C10VEC, the effect of the current on the relative wavelength was computed with the method of Guo (2002). For this, block-averaged horizontal velocities were decomposed in a component in direction of wave propagation and one perpendicular to that. On average, the root mean square error in wave length when neglecting currents was RMSE=0.3 m (3%).

Lastly, applying a linear sea surface reconstruction occurs under the assumption of a random-phase sea state, thereby ignoring the effects of non-linear processes (e.g. triad- and quadruplet wave-wave interactions) that phase couple sea-swell frequencies to higher harmonics that do not adhere to the linear transfer function (e.g. Martins et al., 2021). This means that potentially, bound wave variance, if present, can get over represented in the reconstructed sea-surface signal. The user should be aware of this if higher-order statistics of the reconstructed sea surface (not part of the published tailored dataset) are computed from the raw data.

**More specific comments:**

**1 – L19: along the lines of my comment above on the introduction, what then makes them different than other sandy beaches? What about winds?**

We restructured the introduction, also in line with the earlier comments. The revised introduction identifies the main aspects of sheltered beaches that sets them apart from exposed sand beaches: 1. Tidal and wind driven currents, 2. Isotropic young sea states, 3. Limited restorative capacity of wave-driven onshore transport. 4. Sediment heterogeneity in the case of nourished sheltered beaches. These are discussed in line L34-51.

**2 – The cross-shore and longshore referencing used is really hard to follow in the paper, as it changes between Figures. I found no information on the coordinate system used (only vertical referencing), and in this regard, maybe the mean coastline orientation could be provided so that users could convert easily between real and local coordinate systems (includent long- and cross-shore directions).**

This could indeed have received some more attention in the manuscript. This information was and remains accounted for in the tailored netcdf's for ADV and ADCP instruments. We have altered the manuscript as follow:

L322-323: Positive onshore direction at this cross-section is 135 degrees counterclockwise from East, and positive alongshore direction is 225 degrees counterclockwise from East.

The local coordinate system that is used for figures was indeed not consistent. Now, we have made sure the figures all use the local coordinate system that is also used in the transects netcdf coordinate "d". Figure 3, 8 and 9 were updated. In Figure 9 we showed changes on transect #0 in the earlier manuscript, this is now updated to transect #1 that overlaps the densely measured cross-section.

3 – The unit of raw pressure was not intuitive (Pa around a mean Pa value relative to NAP). At this point, they could be provided directly in m (hydrostatic) relative to NAP or please clarify somewhere.

We agree that the description of the unit Pa+NAP might indeed not be the most intuitive. Therefore we have changed the unit of the pressure on Quality Controlled and tailored timeseries for the pressure sensors to m+NAP so that p indeed represents hydrostatic surface elevation in m referenced to NAP. For the ADV and ADCP instruments, this was already the case for the variable eta. To align ADV/ADCP and pressure sensors, we renamed the variable eta to p, with unit m+NAP on the revised dataset.

**4 – L214: why only resticting to such a narrow frequency band? First, it becomes really sensitive to the definition of the peak frequency, which is clearly hard to define in such wave conditions. Second, the bottom pressure "contains" what reaches the bottom in terms of non-linearities so that including frequencies up to the noise level will be much more representative of wave non-linearities.**

This chosen frequency band, scaling with the peak frequency, was chosen in anticipation of a following piece of work in which we focus on sea-swell wave non-linearity at this and another sheltered site compared to more exposed sites. In that analysis, we observed phase coupling of pressure variance at the peak frequency with lower harmonics in our observations by analysis of bispectra. This bound variance had little influence on spectral wave statistics in the band [0.05, 1] Hz, but was observed to affect the velocity skewness, see e.g. the attached Figure 4 that shows the real and imaginary part of the bispectrum of September 13[th] 12:30 (corresponding to the same time period of the bicoherence plot the reviewer plotted). At more

exposed coasts, this bound variance is identified as infragravity and gets excluded in the integration between [0.05, 1] Hz. Given our short-crested waves, they are observed above 0.05 Hz and hence are included in the integration. This is what motivated us to adjust the frequency band adaptively.

Although we are eager to share a more in depth analysis of the impact of the frequency range we believe this data descriptor manuscript is not the right place to extensively describe this finding. Therefore, we followed the reviewers suggestion to publish velocity skewness and asymmetry in this dataset using the more conventional frequency band [0.05, 1]. With that the statistics are in line with standard processing protocols (e.g. Ruessink et al. 2012, Rocha et al. 2017). A more in-depth discussion on phase coupling in these short-crested waves is saved for another manuscript.

Mentioning of this in the manuscript is modified:

L250 and L268: in the frequency band [0.05-1] Hz

**5 – Sections 8.2 and 8.3 feel light. There is no mention of Figure 9 in section 8.2.**

We expanded the paragraphs 8.2 a bit more to discuss the changes observed in Figure 9 and fixed the missing reference to Figure 9. Likewise, we expanded section 8.3. This section of the article was however only meant to compactly visualize the type of data available, not to analyse the observed signal in-depth.

**6 – All data files have different time origins. Is this the best choice for processing multiples files?**

The reviewer makes a good point. Writing to netcdf was semi-automatically taken care of by the analysis tools we used (python/xarray) on the fly, and the time origin was therefore not a thought over choice. It makes sense indeed to take a stable time origin for ease of analysis using different tools. We have modified the time origin in the revised dataset for all instruments to 2021-09-01 00:00:00 and set the time axis in seconds from this origin.

**7 – The table indicates 30 min bursts for the ADV L2C10 while the data actually suggests it is 10 min. Please verify that the figures provided in the table are correct.**

ADV's recorded raw data in bursts of 30 minutes indeed, as tabulated in Table 1. The reader was left confused because the decision to analyse in 10 minute blocks was only mentioned very briefly in Section 6. The explanation of this discrepancy is now more explicitly stated in Section 6:

L194-198: For all instruments apart from the ADCP, it was decided to perform analysis on 10 minute data blocks instead of the full 30 minute bursts. At this site with short crested waves, 10 minute blocks are sufficiently long for mean spectral statistics in the sea-swell band and 10 minute blocks increase the total duration of valid measurement blocks with submerged instruments. Therefore, the raw data folder consists of data structured in 30 minute bursts, and qc and tailored data folders contain data structured into 10 minute bursts.

**8 – In relation to #7: the Table indicates that the sampling frequency of L2C10SOLO is 8 Hz, while the data file indicates 10 Hz. By comparing with the pressure from the ADV, it**

clearly is 8 Hz. Please verify that all netcdf files include the correct information. As reviewers, we cannot verify every individual files…

We thank the reviewer for finding this error, the sampling frequency of the SOLO pressure sensors was 8 Hz indeed. In a last modification of the processing scripts the sampling frequency of the SOLO sensors was mistakenly set to that of the OSSI sensors. We have corrected this now on the SOLO dataset, and this indeed also affects the wave statistics reported at the SOLO tailored datasets. All other instrument files were checked after identification of this error, and were found to be fine.

**9 – I felt that more information could be provided within the netcdf files, and fields could be more descriptive in general.**

We have reviewed the attributes of variables and added units and long names to some variables where these were mistakenly missing. Thanks for pointing this out to us. This was for example the case for the coordinate N, the block local time, and the vertical coordinate z for the ADCP datasets. On top of that, we have added the campaign summary that is also part of the 4TU overview page to the general dataset attributes for each file in the hydrodynamics dataset.

[Figure]

Figure 1 Effect of capping of transfer function on computed wave statistics for the four moments in time analysed by the reviewer. The left panel shows the transfer function with cap and without as a function of frequency, and the reconstructed variance density with and without cap. The right panels show the effect of the cap on the transfer function on the sensitivity of the cut-off frequency

[Figure]

*Figure 2 Effect of correcting for ambient current on wave length at ADV L2C10VEC. 2D histogram of current corrected wave length versus wave length without current correction. Dashed lines indicate mean value of wave length and current corrected wave length of entire timeseries.*

[Figure]

*Figure 3 Effect of correcting for ambient current on wave length at ADV L2C10VEC. 2D histogram of current corrected wave length versus wave length without current correction. Dashed lines indicate mean value of wave length and current corrected wave length of entire timeseries.*

[Figure]

*Figure 4 Bispectrum of calm burst on September 13th 12:30. Power spectrum in top panel, Imaginary and Real part of the Bispectrum plotted below. Legend of colorbar mentions statistical significance threshold of bicoherence (Bicoherence not shown here).*

---

## Author Response (AR2)

**The following fixes as suggested by the editor were made to the bibliography:**

Sonu 1972: added page numbers

Colosimo 2023: added volume

Cooper 2007: added special issue number and page numbers

Guo 2002: added volume and page numbers

Hegermiller 2022: added electronic page numbers

Martins 2021: added volume and page numbers

**Moreover, we fixed three more mistakes in the bibliography:**

Donelan 1985: fixed journal name

Pearson 2019: changed preprint to published version

Van der Lugt 2023: fixed the doi from private view to publicly available one